# Understanding and Improving Layer Normalization

**Jingjing Xu[1], Xu Sun[1,2]***, **Zhiyuan Zhang[1], Guangxiang Zhao[2], Junyang Lin[1]**
[1] MOE Key Lab of Computational Linguistics, School of EECS, Peking University
[2] Center for Data Science, Peking University
{jingjingxu,xusun,zzy1210,zhaoguangxiang,linjunyang}@pku.edu.cn

## Abstract

Layer normalization (LayerNorm) is a technique to normalize the distributions of intermediate layers. It enables smoother gradients, faster training, and better generalization accuracy. However, it is still unclear where the effectiveness stems from. In this paper, our main contribution is to take a step further in understanding LayerNorm. Many of previous studies believe that the success of LayerNorm comes from forward normalization. Unlike them, we find that the derivatives of the mean and variance are more important than forward normalization by re-centering and re-scaling backward gradients. Furthermore, we find that the parameters of LayerNorm, including the bias and gain, increase the risk of over-fitting and do not work in most cases. Experiments show that a simple version of LayerNorm (LayerNorm-simple) without the bias and gain outperforms LayerNorm on four datasets. It obtains the state-of-the-art performance on En-Vi machine translation. To address the over-fitting problem, we propose a new normalization method, Adaptive Normalization (AdaNorm), by replacing the bias and gain with a new transformation function. Experiments show that AdaNorm demonstrates better results than LayerNorm on seven out of eight datasets.

## 1   Introduction

Neural network training has long been a focus in Deep Learning research area. One of the prominent progress is the application of normalization methods. Initially, Ioffe and Szegedy [2015] introduce the concept of normalizing layers with the proposed Batch Normalization (BatchNorm). It is widely believed that by controlling the mean and variance of layer inputs across mini-batches, BatchNorm stabilizes the distribution and improves training efficiency. Following this work, Lei Ba et al. [2016] point out its limitation in Recurrent Neural Networks (RNN) and propose Layer Normalization (LayerNorm) that is performed across the neurons in a layer. LayerNorm is adaptive to RNN and self-attention-based models. A typical example is its application in the state-of-the-art framework, Transformer [Vaswani et al., 2017]. LayerNorm enables faster training of Transformer and is irreplaceable in this framework.

Despite its great success, it is still unclear why LayerNorm is so effective. The widely accepted explanation is that forward normalization brings distribution stability [Ioffe and Szegedy, 2015, Lei Ba et al., 2016]. Recent studies show that the effects of BatchNorm are not related to the stability of input distribution [Zhang et al., 2017, Santurkar et al., 2018]. They also propose that the reason why BatchNorm is effective is that normalization smooths the optimization landscape. However, it is still unclear whether these theories can explain the success of LayerNorm.

The main contribution of this paper is to explore how LayerNorm works. Through a series of analyses, we find that the derivatives of the mean and variance are important by re-centering and re-scaling

---

backward gradients. Furthermore, it is beyond our expectation that the bias and gain do not work in most cases. The details of our findings are illustrated below.

**The derivatives of the mean and variance are more important to LayerNorm than forward normalization.** Many of the previous studies believe that the forward normalization is the only decisive factor to LayerNorm. It makes the input distribution more stable, thus brings better convergence. Unlike them, our experimental results show that forward normalization has little to do with the effectiveness and the derivatives of the mean and variance play a significant role in LayerNorm. To illustrate how these derivatives work, we propose DetachNorm, which adds an additional detaching operation to LayerNorm to change the mean and variance from variables to constants. It preserves the re-centering and re-scaling fact but cuts off the derivative of the mean and variance with respect to the input. DetachNorm performs worse than LayerNorm on six out of eight datasets. This proves that the derivatives of the mean and variance are useful to LayerNorm. Furthermore, to investigate the reason for the above observation, we analyze the gradients in LayerNorm and DetachNorm, and find that the derivatives of means re-center gradients and the derivatives of variances re-scale gradients.

**The parameters of LayerNorm, including the bias and gain, increase the risk of over-fitting and do not work in most cases.** The bias and gain are applied for affine transformation on normalized vectors. They are expected to enhance the expressive power by re-shaping the distribution. To evaluate their effects on results, we build a simple version of LayerNorm (LayerNorm-simple) by removing the bias and gain. Our experimental results show that LayerNorm-simple achieves better results than LayerNorm on four datasets. It even achieves the state-of-the-art performance on En-Vi machine translation. By comparing loss curves of LayerNorm with and without the bias and gain, we find that the bias and gain cause over-fitting. We speculate the reason of over-fitting is mainly that the bias and gain are learned from the training set and cannot adjust themself towards different input distributions when testing.

Motivated by this assumption, we propose a novel normalization method, Adaptive Normalization (AdaNorm). AdaNorm replaces the bias and gain with a new transformation function. This function adaptively adjusts scaling weights based on input values. We evaluate AdaNorm and LayerNorm on eight datasets, covering tasks of machine translation, language modeling, text classification, image classification, and dependency parsing. Results show that AdaNorm achieves better results on seven datasets.

## 2 Preliminaries

In this section, we first review the algorithm of LayerNorm and then introduce the datasets and models used in the following analysis sections.

### 2.1 LayerNorm Algorithm

Let $\mathbf{x} = (x_1, x_2, \ldots, x_H)$ be the vector representation of an input of size $H$ to normalization layers. LayerNorm re-centers and re-scales input $\mathbf{x}$ as

$$\mathbf{h} = \mathbf{g} \odot N(\mathbf{x}) + \mathbf{b}, \quad N(\mathbf{x}) = \frac{\mathbf{x} - \mu}{\sigma}, \quad \mu = \frac{1}{H}\sum_{i=1}^{H} x_i, \quad \sigma = \sqrt{\frac{1}{H}\sum_{i=1}^{H}(x_i - \mu)^2} \quad (1)$$

where $\mathbf{h}$ is the output of a LayerNorm layer. $\odot$ is a dot production operation. $\mu$ and $\sigma$ are the mean and standard deviation of input. Bias $\mathbf{b}$ and gain $\mathbf{g}$ are parameters with the same dimension $H$.

### 2.2 Experimental Setup

To investigate how LayerNorm works, we conduct a series of experiments in this paper. Since LayerNorm is a default setting in Transformer [Vaswani et al., 2017] and Transformer-XL [Dai et al., 2019], which have shown state-of-the-art results on a variety of tasks (e.g., machine translation), we primarily consider normalization on Transformer and Transformer-XL networks. Also, to avoid the impact of model architecture, we evaluate the effects of normalization on feed-forward neural networks and convolutional neural networks. Here list the datasets and models. More details can be found at the arxiv version.

**Machine translation** includes three widely-used datasets, WMT English-German (En-De), IWSLT 14 German-English (De-En) [Cettolo et al., 2014] and IWSLT 15 English-Vietnamese (En-Vi) [Cettolo et al., 2015]. For all dataset, we use the setting of PreNorm where normalization is applied before each layer. We re-implement Transformer with the released code of Fairseq [Ott et al., 2019][2]. The evaluation metric is BLEU [Papineni et al., 2002].

For En-De dataset, we use the same dataset splits and the same compound splitting following previous work [Vaswani et al., 2017]. BPE is used to get vocabularies. We use the shared embedding setting and the vocabulary size is 32,765. We use "transformer_wmt_en_de_big_t2t" as our basic model. The dropout rate is 0.3. The learning rate is 0.001. The training batch size is 4,096 tokens. We use optimizer Adam with $\beta_1 = 0.9$ and $\beta_2 = 0.98$. The number of warmup steps is 4K.

The De-En dataset is provided by the IWSLT 2014 Evaluation Campaign. We use the same dataset splits following previous work [Ott et al., 2019, Ranzato et al., 2016, Wiseman and Rush, 2016]. It contains 153K sentences for training, 7K sentences for validation, and 7K sentences for testing. BPE is used to get vocabularies. We use the shared embedding setting and the vocabulary size is 10,149. We use "transformer_iwslt_de_en" as our basic model. The dropout rate is 0.3. The attention dropout rate is 0.1. The activation dropout is 0.1. The initialization learning rate is 1e-07 and the learning rate is 0.0015. The training batch size is 4,096 tokens. We update gradients for every 2 steps. The number of warmup steps is 8K.

The En-Vi dataset contains 133K training sentence pairs provided by the IWSLT 2015 Evaluation Campaign. We use TED tst2012 (1,553 sentences) as the validation set and TED tst2013 (1,268 sentences) as the test set. BPE is used to get input and output vocabularies. The English and Vietnamese vocabulary sizes are 7,669 and 6,669 respectively. The dropout rate is 0.1. The learning rate is 0.001. The training batch size is 4,096 tokens. The number of warmup steps is 8K. We use "transformer_wmt_en_de" as our basic model. We use optimizer Adam with $\beta_1 = 0.9$ and $\beta_2 = 0.98$.

**Language modeling** includes a large dataset, Enwiki8[3] that contains 100M bytes of unprocessed Wikipedia text. We implement a 12-layer Transformer-XL model. The dimension of each layer is 512. Multi-head attention contains 8 heads and the dimension of each head is 64. The dropout rate is 0.1. The batch size is 22. We use optimizer Adam with a learning rate 0.00025. We use the average number of Bits-Per-Character (BPC) as the evaluation metric [Al-Rfou et al., 2018, Dai et al., 2019].

**Text classification** includes two sentence classification datasets: RT [Pang and Lee, 2005], and SST5 [Socher et al., 2013]. RT is a binary sentiment classification dataset from online movie reviews. We randomly divide all examples into 8,608 for training, 964 for validation, and 1,089 for testing. SST5 is a single-sentence classification dataset built on movie reviews. We run experiments on a five label set. We build a Transformer model with a 4-layer encoder. The batch size is 4,096 tokens. The word embedding dimension is 128 and the hidden dimension is 128. The dropout rate is 0.2. We use optimizer Adam with $\beta_1 = 0.9$, $\beta_2 = 0.998$. Normalization is applied before each layer. Accuracy is the evaluation metric.

**Image classification** includes a widely-used dataset, MNIST [LeCun et al., 1998]. It consists of 55,000 training images, 5,000 validation images, and additional 10,000 testing images. We implement a 3-layer convolutional neural network for classification. The first 2D-convolution layer has 1 in-channel, 20 out-channels. The second 2D-convolution layer has 20 in-channels, 50 out-channels. We flatten the output of the second 2D-convolution layer and send it to a linear layer. The batch size is 32. We use optimizer Adam with a learning rate of 0.001. We apply LayerNorm before the activation in every linear layer. We train the model for 20 epochs. Normalization is applied before each layer. Accuracy is the evaluation metric.

**Dependency parsing** includes a dataset, English Penn TreeBank (PTB) [Marcus et al., 1993]. We follow the standard split of the corpus with sections 2-21 as the training set (39,832 sentences, 1,900,056 transition examples), section 22 as the validation set (1,700 sentences, 80,234 transition examples), and section 23 as the testing set (2,416 sentences, 113,368 transition examples). We implement a MLP-based parser following the work [Chen and Manning, 2014]. The dimension of the hidden state is 512, the batch size is $1,024$, the dropout rate is 0.2. We use optimizer Adam and initialize the learning rate to 0.001. We apply normalization before activation in every linear layer.

Following the work [Chen and Manning, 2014], we use Unlabeled Attachment Score (UAS) as the evaluation metric.

## 3 Understanding LayerNorm

To investigate how LayerNorm facilitates training, we conduct ablation studies to observe each part's contribution to the performance. In this section, we analyse the effects of the bias and gain, forward normalization, and backward normalization.

Table 1: The bias and gain do not work on six out of eight datasets. "w/o Norm" is a naive model without LayerNorm. "LayerNorm-simple" is a variant of LayerNorm that drops the bias and gain. "(+)" means higher is better. "(-)" means lower is better.

| Models | Machine Translation | | | Language Modeling | Classification | | | Parsing |
|---|---|---|---|---|---|---|---|---|
| | En-De(+) | De-En(+) | En-Vi(+) | Enwiki8(-) | RT(+) | SST5(+) | MNIST(+) | PTB(+) |
| Model Layers | 12 | 12 | 12 | 12 | 4 | 4 | 3 | 3 |
| w/o Norm | Diverge | 34.0 | 28.4 | **1.04** | 76.85 | 38.55 | **99.14** | 88.31 |
| LayerNorm | 28.3 | **35.5** | 31.2 | 1.07 | **77.21** | 39.23 | 99.13 | 89.12 |
| LayerNorm-simple | **28.4** | **35.5** | **31.6** | 1.07 | 76.66 | **40.54** | 99.09 | **89.19** |

### 3.1 The Effect of the Bias and Gain in LayerNorm

The bias and gain do not work in most cases. From Table 1, it can be found that LayerNorm is an effective approach. It brings large performance improvements on six out of eight datasets compared with the naive baseline without LayerNorm ("w/o Norm"). By comparing LayerNorm and LayerNorm-simple, we find that dropping the bias and gain ("LayerNorm-simple") does not decrease the performance on six datasets. Surprisingly, LayerNorm-simple outperforms LayerNorm on four datasets, even with a 0.4 BLEU improvement on En-Vi and a 1.31 ACC improvement on SST-5. Also, it needs to notice that 31.6 achieved by LayerNorm-simple is the state-of-the-art result on En-Vi machine translation.

Furthermore, we find that the bias and gain increase the risk of over-fitting. Initially, considering that input information may be lost when normalizing input distributions, the bias and gain are designed for affine transformation on normalized vectors to enhance the expressive power. However, since the bias and gain are learned from the training set and they ignore the input distributions of the testing data, the risk of over-fitting may increase in LayerNorm. It is verified by convergence curves in Figure 1. LayerNorm achieves lower training loss (or BPC) but higher validation loss (or BPC) than LayerNorm-simple on En-Vi, Enwiki8. These results indicate that current affine transformation mechanism has a potential risk of over-fitting and needs to be further improved.

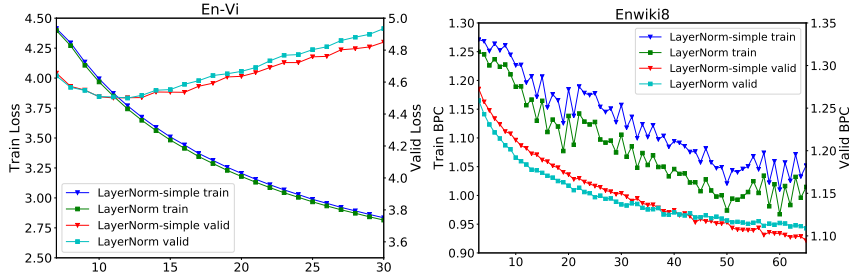

Figure 1: Convergence curves of LayerNorm and LayerNorm-simple on En-Vi, Enwiki8. Lower is better. The bias and gain increase the risk of over-fitting.

### 3.2 The Effect of Forward Normalization

For easier analysis, we only consider LayerNorm without the bias and gain here. Let $\mathbf{y} = (y_1, y_2, \ldots, y_H)$ be the normalized vector, the calculation process of LayerNorm without the bias and

Table 2: The derivatives of the mean and variance matter. "w/o Norm" is the naive model without normalization. "DetachNorm" is a variant of "LayerNorm-simple". It detaches the derivatives of the mean and variance. "(+)" means higher is better. "(-)" means lower is better. The top table shows the effect of forward normalization. The bottom table shows the effect of the derivatives of the mean and variance.

| Models | Machine Translation | | | Language Modeling | Classification | | | Parsing |
|---|---|---|---|---|---|---|---|---|
| | En-De | De-En(+) | En-Vi(+) | Enwiki8(-) | RT(+) | SST5(+) | MNIST(+) | PTB(+) |
| Model Layers | 12 | 12 | 12 | 12 | 4 | 4 | 3 | 3 |
| w/o Norm | Diverge | **34.0** | **28.4** | **1.04** | **76.85** | 38.55 | **99.14** | 88.31 |
| DetachNorm | Diverge | 33.9 | 27.7 | 1.12 | 76.40 | **40.04** | 99.10 | **89.79** |
| Improvement | – | -0.1 | -0.7 | -0.08 | -0.45 | 1.49 | -0.04 | **1.48** |
| Models | Machine Translation | | | Language Modeling | Classification | | | Parsing |
| | En-De | De-En(+) | En-Vi(+) | Enwiki8(-) | RT(+) | SST5(+) | MNIST(+) | PTB(+) |
| Model Layers | 12 | 12 | 12 | 12 | 4 | 4 | 3 | 3 |
| DetachNorm | Diverge | 33.9 | 27.7 | 1.12 | 76.40 | 40.04 | **99.10** | 89.79 |
| LayerNorm-simple | **28.4** | **35.5** | **31.6** | **1.07** | **76.66** | 40.54 | 99.09 | 89.19 |
| Improvement | – | **1.6** | **3.9** | **0.05** | **0.26** | **0.50** | -0.01 | -0.60 |

gain can be written as

$$\mathbf{y} = \frac{\mathbf{x} - \mu}{\sigma}, \quad \mu = \frac{1}{H} \sum_{i=1}^{H} x_i, \quad \sigma = \sqrt{\frac{1}{H} \sum_{i=1}^{H} (x_i - \mu)^2} \qquad (2)$$

where $\mathbf{x} = (x_1, x_2, \ldots, x_H)$ is the input vector and $H$ is the dimension of $\mathbf{x}$. $\mu$ and $\sigma$ are the mean and standard deviation of $x_1, x_2, \ldots, x_H$. Then, suppose $\bar{y}$ and $D_y$ are the mean and variance of $y_1, y_2, \ldots, y_H$. It is easy to verify

$$\bar{y} = \frac{1}{H} \sum_{i=1}^{H} y_i = \frac{1}{H} \sum_{i=1}^{H} \frac{(x_i - \mu)}{\sigma} = 0, \quad D_y = \frac{1}{H} \sum_{i=1}^{H} \frac{(x_i - \mu)^2}{\sigma^2} = 1. \qquad (3)$$

Eq. (3) shows that normalization re-centers and re-scales input vector $\mathbf{x}$. By now, a widely accepted belief is that the effectiveness of LayerNorm comes from steady layer distributions brought by forward normalization [Lei Ba et al., 2016]. To evaluate whether forward normalization explains the effectiveness of LayerNorm, we need to separate the effect on forward layer inputs and that on backward gradients. In this paper, we design a new method, called DetachNorm. The difference between LayerNorm and DetachNorm is that DetachNorm detaches the derivatives of the mean and variance[4]. Detaching derivatives means treating the mean and variance as changeable constants, rather than variables, which do not require gradients in backward propagation. The calculation of DetachNorm can be written as

$$\mathbf{y} = \frac{\mathbf{x} - \hat{\mu}}{\hat{\sigma}}, \quad \hat{\mu} = \theta(\mu), \quad \hat{\sigma} = \theta(\sigma) \qquad (4)$$

where $\mu$ and $\sigma$ are the mean and standard deviation of input $\mathbf{x}$, as calculated in Eq. (2). The function $\theta(\cdot)$ can be seen as a special copy function, which copies the values of $\mu$ and $\sigma$ into constants $\hat{\mu}$ and $\hat{\sigma}$. In all, DetachNorm keeps the same forward normalization fact as LayerNorm does, but cuts offs the derivatives of the mean and variance.

Since DetachNorm keeps the same re-centering and re-scaling way in forward propagation as LayerNorm-simple does, the gap between DetachNorm and "w/o Norm" shows the effect of forward normalization. As we can see, DetachNorm perform worse than "w/o Norm", showing that forward normalization has little to do with the success of LayerNorm.

Furthermore, the only difference between DetachNorm and LayerNorm-simple lies in that Detach-Norm detaches the derivatives of the mean and variance. As shown in Table 2, DetachNorm performs

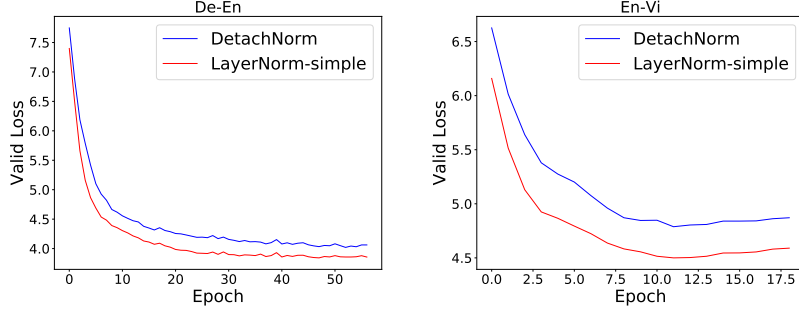

Figure 2: Convergence curves of LayerNorm-simple and DetachNorm on two translation datasets.

worse than LayerNorm-simple on six datasets. It is mainly because that DetachNorm converges to much worse local optima compared with LayerNorm-simple, as shown in Figure 2. The gap between DetachNorm and LayerNorm-simple shows the effectiveness of the derivatives of the mean and variance. By comparing the achieved improvements, we find that the derivatives of the mean and variance bring higher improvements than forward normalization does.

These results demonstrate that the derivatives of the mean and variance play a significant role. In addition, the extremely worse results of DetachNorm on En-De, De-En and En-Vi indicate that the derivatives of the mean and variance may be more important for deeper models. In the following section, we will give a detailed analysis of why and how the derivatives of the mean and variance contribute to the performance.

### 3.3 The Effect of the Derivatives of the Mean and Variance

To understand how the derivatives of the mean and variance work, we analyze the gradients of LayerNorm-simple and DetachNorm. According to the chain rule, the gradient of $\mathbf{x}$ is[5]

$$\frac{\partial \ell}{\partial \mathbf{x}} \leftarrow \frac{d\mathbf{y}}{d\mathbf{x}} \frac{\partial \ell}{\partial \mathbf{y}} \tag{5}$$

where $\ell$ is the loss function, $\mathbf{x}$ is the input vector and $\mathbf{y}$ is the normalized vector. We here analyze the effect of detaching the derivatives of the mean and variance on backward gradients. Our results are summarized in the following theorem, whose proof is listed in the Appendix of the arxiv version.

**Theorem 1.** *Given $\frac{\partial \ell}{\partial \mathbf{y}} = (g_1, g_2, ..., g_H)^T$, let $\bar{g}$ and $D_g$ be the mean and variance of $g_1, g_2, ..., g_H$. For the case of detaching the derivatives of $\mu$ and $\sigma$, suppose $\frac{\partial \ell}{\partial \boldsymbol{x}} = (a_1, a_2, ..., a_H)^T$ is the gradient of $\boldsymbol{x}$ with mean $\bar{a}$ and variance $D_a$. We have $\bar{a} = \bar{g}/\sigma$ and $D_a = D_g/(\sigma^2)$.*

*(1) For the case of standard LayerNorm-simple, suppose $\frac{\partial \ell}{\partial \boldsymbol{x}} = (b_1, b_2, ..., b_H)^T$ is the gradient of $\boldsymbol{x}$ with mean $\bar{b}$ and variance $D_b$.*

*We have $\bar{b} = 0$ and $D_b \leq D_g/(\sigma^2)$.*

*(2) For the case of detaching the derivative of $\mu$, suppose $\frac{\partial \ell}{\partial \boldsymbol{x}} = (c_1, c_2, ..., c_H)^T$ is the gradient of $\boldsymbol{x}$ with mean $\bar{c}$ and variance $D_c$.*

*We have $\bar{c} = \bar{g}/\sigma$ and $D_c \leq D_g/(\sigma^2)$.*

*(3) For the case of detaching the derivative of $\sigma$, suppose $\frac{\partial \ell}{\partial \boldsymbol{x}} = (d_1, d_2, ..., d_H)^T$ is the gradient of $\boldsymbol{x}$ with mean $\bar{d}$ and variance $D_d$.*

*We have $\bar{d} = 0$ and $D_c = D_g/(\sigma^2)$.*

By comparing the case of detaching the derivative of $\mu$ with that of LayerNorm-simple in Theorem 1, we find that the derivative of $\mu$ re-centers $\frac{\partial \ell}{\partial \mathbf{x}}$ to zero. By comparing the case of detaching the

derivative of $\sigma$ with of LayerNorm-simple, we find that the derivative of $\sigma$ reduces the variance of $\frac{\partial \ell}{\partial \mathbf{x}}$, which can be seen a kind of re-scaling. We refer to gradient re-centering and re-scaling as gradient normalization.

To further evaluate the effect of gradient normalization on model performance, we test the derivatives of the mean and variance separately. Table 3 shows that detaching the derivative of variance decreases the performance significantly on deeper networks. Therefore, it is necessary to control the variance of gradients for deeper networks.

In conclusion, LayerNorm normalizes forward layer inputs and backward gradients. The derivatives of the mean and variance play more important roles than forward normalization in LayerNorm. Furthermore, unlike previous work [Santurkar et al., 2018] only noticing that normalization smooths gradients, this paper provides deeper insight about how normalization impacts backward gradients.

Table 3: The derivative of variance is more important than that of mean for deeper networks. "(+)" means higher is better. "(-)" means lower is better.

| Models | Machine Translation | | | Language Model | Classification | | | Parsing |
|---|---|---|---|---|---|---|---|---|
| | En-De(+) | De-En(+) | En-Vi(+) | Enwiki8(-) | RT(+) | SST5(+) | MNIST(+) | PTB(+) |
| Model Layers | 12 | 12 | 12 | 12 | 4 | 4 | 3 | 3 |
| LayerNorm-simple | **28.4** | 35.5 | **31.6** | **1.07** | 76.66 | 40.54 | 99.09 | 89.19 |
| Detach Mean | 28.3 | **35.6** | 31.3 | **1.07** | 75.02 | 40.99 | **99.25** | 89.45 |
| Detach Variance | Diverge | 34.2 | 29.8 | 1.10 | **77.04** | **41.74** | 99.10 | **89.80** |

## 4 AdaNorm

AdaNorm adopts a new transformation function which can adaptively control scaling weights towards different inputs.[6]

### 4.1 AdaNorm Algorithm

Formally, let $\mathbf{y} = N(\mathbf{x}) = (\mathbf{x} - \mu)/\sigma$ be the normalized vector where $\mu$ and $\sigma$ are the mean and variance of the input $\mathbf{x} = (x_1, x_2, \ldots, x_H)$. We use $\phi(\mathbf{y})$, a function with respect to input $\mathbf{x}$, to replace the bias and gain with the following equation:

$$\mathbf{z} = \phi(\mathbf{y}) \odot \mathbf{y} = \phi(N(\mathbf{x})) \odot N(\mathbf{x}) \tag{6}$$

where $\mathbf{z} = (z_1, z_2, \ldots, z_H)$ is the output of AdaNorm and $\odot$ is a dot product operation. Unlike the bias and gain being fixed in LayerNorm, $\phi(\mathbf{y})$ can adaptively adjust scaling weights based on inputs. To keep the stability of training, we expect that $\phi(\cdot)$ has some features. First, $\phi(\cdot)$ must be differentiable. Second, we expect that the average scaling weight is fixed, namely the average of $\phi(\mathbf{y})$ is a constant $C$ where $C > 0$. Third, we expect that the average of $\mathbf{z}$ is bounded, which can avoid the problem of exploding loss. Namely, we require that there exists a constant $M$ such that $|\frac{1}{H} \sum\limits_{i=1}^{H} z_i| < M$. Theorem 2 proves that there exists a unique solution which can satisfy these requirements. The proof is listed in the Appendix of the arxiv version.

**Theorem 2.** *Suppose $\phi(y_i)$ is derivable, $\forall \mathbf{y}$, $\frac{1}{H} \sum\limits_{i=1}^{H} \phi(y_i) = C > 0$, and $\exists M, s.t. |\frac{1}{H} \sum\limits_{i=1}^{H} z_i| < M$ ($M > 0$), where $H$ is the hidden size. There exists only one solution:*

$$\phi(y_i) = C(1 - ky_i)$$

*which can satisfy these requirements.*

Since $1 - ky_i < 0$ will undesirably change the direction of vector, we expect that $\phi(y_i) > 0$ holds, which means $y_i < 1/k$ must hold. Due to the symmetry of $y_i$, $|y_i| < 1/k$ is required to hold too.

Based on Chebyshev's Inequality, we have

$$P(|y_i| < 1/k) = P(|y_i - E(y_i)| < 1/k) \geq 1 - \frac{D_y}{(1/k)^2} = 1 - k^2 D_y \qquad (7)$$

where $D_y$ is the variance of $\mathbf{y} = (y_1, y_2, \ldots, y_H)$ and $H$ is the dimension of $\mathbf{y}$. Based on Eq. (3), we can verify $D_y = 1$. If we expect that $|y_i| < 1/k$ holds with a probability higher than $99\%$, $k = 1/10$ should be choose based on Eq. (7). Namely, we choose

$$\phi(y_i) = C(1 - \frac{y_i}{10}). \qquad (8)$$

Given an input vector $\mathbf{x}$, the complete calculation process of AdaNorm is

$$\mathbf{z} = C(1 - k\mathbf{y}) \odot \mathbf{y}, \quad \mathbf{y} = \frac{\mathbf{x} - \mu}{\sigma}, \quad \mu = \frac{1}{H} \sum_{i=1}^{H} x_i, \quad \sigma = \sqrt{\frac{1}{H} \sum_{i=1}^{H} (x_i - \mu)^2} \qquad (9)$$

where $C$ is a hyper-parameter. $\odot$ is a dot product operation. $k$ is recommended to set as $1/10$. To prevent the introduced term $C(1 - k\mathbf{y})$ dismissing the feature of gradient re-centering and re-scaling, we detach the gradient of $C(1 - k\mathbf{y})$ and only treat it as a changeable constant in implementation.

Table 4: Results of LayerNorm and AdaNorm. "(+)" means higher is better. "(-)" means lower is better. AdaNorm outperforms LayerNorm on seven datasets.

| Models | Machine Translation | | | Language Model | Classification | | | Parsing |
|---|---|---|---|---|---|---|---|---|
| | En-De(+) | De-En(+) | En-Vi(+) | Enwiki8(-) | RT(+) | SST5(+) | MNIST(+) | PTB(+) |
| w/o Norm | Diverge | 34.0 | 28.4 | **1.04** | 76.85 | 38.55 | 99.14 | 88.31 |
| LayerNorm | 28.3 | 35.5 | 31.2 | 1.07 | 77.21 | 39.23 | 99.13 | 89.12 |
| LayerNorm-simple | 28.4 | 35.5 | **31.6** | 1.07 | 76.66 | **40.54** | 99.09 | 89.19 |
| AdaNorm | **28.5** | **35.6** | 31.4 | 1.07 | **77.50** | **40.54** | **99.35** | **89.23** |

## 4.2 Comparison between AdaNorm and LayerNorm

The comparison between LayerNorm and AdaNorm is shown in Table 4.[7] AdaNorm outperforms LayerNorm on seven datasets, with 0.2 BLEU on En-De, 0.1 BLEU on De-En, 0.2 BLEU on En-Vi, 0.29 ACC on RT, 1.31 ACC on SST, 0.22 ACC on MNIST, and 0.11 UAC on PTB. Unlike LayerNorm-simple only performing well on bigger models, AdaNorm achieves more balanced results. Figure 3 shows the loss curves of LayerNorm and AdaNorm on the validation set of En-Vi, PTB, and De-En. Compared to AdaNorm, LayerNorm has lower training loss but higher validation loss. Lower validation loss proves that AdaNorm has better convergence.

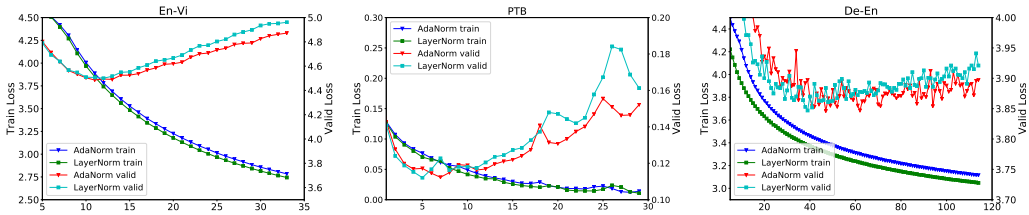

Figure 3: Loss curves of LayerNorm and AdaNorm on En-Vi, PTB, and De-En.

## 5 Related Work

Deep neural networks have outperformed shallow models in a variety of fields, such as natural language processing [Sutskever et al., 2014, Bahdanau et al., 2015, Devlin et al., 2018], computer vision [He et al., 2016, Huang et al., 2017], etc. The improvement mainly comes from the stronger

expressive power of deep layers. However, with the increase of depth, the network training process becomes complicated and requires advanced architectural techniques. One of the important techniques of such advances is normalization.

Currently, it is widely accepted that normalization layers assist training by smoothing gradients, enabling large learning rates, accelerating convergence, and improving generalization results [Zhang et al., 2019]. First introduced by Ioffe and Szegedy [2015], BatchNorm fixes layer distributions to reduce ICS (Internal Covariate Shift), a phenomenon that the upper layers need to continuously adapt to the new distributions of lower layers. Following this work, several normalization methods have been proposed, like instance normalization [Ulyanov et al., 2016] and group normalization [Wu and He, 2018]. In addition, there are several studies exploring better activation functions [Klambauer et al., 2017] or initialization methods [Zhang et al., 2019] to avoid the dependency on normalization layers.

LayerNorm is proposed to expand BatchNorm into RNN. LayerNorm normalizes the mean and variance of all summed inputs to the neurons in one layer. Unlike BatchNorm that depends on the size of mini-batch, LayerNorm has fewer limitations. LayerNorm is adaptive to RNN and self-attention-based models. It has been applied to the state-of-the-art frameworks such as Transformer [Vaswani et al., 2017], BERT [Devlin et al., 2018], and Transformer-XL [Dai et al., 2019]. LayerNorm brings better performance and is irreplaceable in these frameworks.

Despite the good performance, it is still unclear how layer normalization works. Ioffe and Szegedy [2015] claim that the effectiveness of BatchNorm comes from reducing ICS. It has been a popular belief about BatchNorm [Santurkar et al., 2018]. However, some recent studies point out that the success of BatchNorm relates to the smoother gradients and has little to do with reducing ICS [Santurkar et al., 2018, Bjorck et al., 2018]. Although these studies provide a pioneering perspective to understand BatchNorm, there still remain some unanswered questions, such as how BatchNorm helps smooth gradients. Also, there are little work studying whether these theories can explain the success of LayerNorm. In this paper, we take a further step to a better understanding of LayerNorm.

# 6 Conclusion

In this paper, we investigate how layer normalization works. Based on a series of experiments and theoretical analysis, we summarize some interesting conclusions. We find that the derivatives of the mean and variance are important to the success of LayerNorm by re-centering and re-scaling backward gradients. Furthermore, experiments show that the bias and gain increase the risk of over-fitting and do not work in most cases. To address this problem, we propose a normalization method AdaNorm. It replaces the bias and gain in LayerNorm with a new adaptive transformation function that can update scaling weights based on input values. Experiments show that AdaNorm outperforms LayerNorm on seven datasets. In the future work, we would like to explore more alternatives to LayerNorm from the perspective of gradient normalization.

# Acknowledgments

We thank all reviewers for providing the thoughtful and constructive suggestions. This work was supported in part by National Natural Science Foundation of China (No. 61673028).

## Footnotes

[2]https://github.com/pytorch/fairseq

[3]http://mattmahoney.net/dc/text.html

[4]In our implementation, we detach the derivative of standard deviation, the square root of variance.

[5]When calculating the gradient, we adopt the denominator layout.

[6]Our code is released at `https://github.com/lancopku/AdaNorm`

[7]For AdaNorm implementation, Kaiming initialization and the setting of prenorm are recommended.

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
