[Supplementary Material · Supplementary Material (1).pdf]

# A  Experimental Settings

## A.1  Neural Machine Translation

For neural machine translation tasks, we re-implement Transformer with the released code of Fairseq [Ott et al., 2019][1].

**IWSLT 2015 English-Vietnamese Translation** It contains 133K training sentence pairs provided by the IWSLT 2015 Evaluation Campaign. Following the pre-processing steps in the work of Raffel et al. [2017], we use TED tst2012 (1,553 sentences) as the validation set and TED tst2013 (1,268 sentences) as the test set. BPE is used to get input and output vocabularies. The English and Vietnamese vocabulary sizes are 7,669 and 6,669 respectively. The dropout rate is 0.1. The learning rate is 0.001. The training batch size is 4,096 tokens. The number of warmup steps is 8K. We use "transformer_wmt_en_de" as our basic model. The setting of PreNorm is adopted. We use optimizer Adam with $\beta_1 = 0.9$ and $\beta_2 = 0.98$. For AdaNorm, the hyper-parameter $C$ is set to 1. We average the last 10 checkpoints for evaluation and set the beam size to 5.

**IWSLT 2014 German-English Translation** It is provided by the IWSLT 2014 Evaluation Campaign. We use the same dataset splits following previous work [Ott et al., 2019, Ranzato et al., 2016, Wiseman and Rush, 2016]. It contains 153K sentences for training, 7K sentences for validation, and 7K sentences for testing. BPE is used to get vocabularies. We use the shared embedding setting and the vocabulary size is 10,149. We use "transformer_iwslt_de_en" as our basic model. The setting of PreNorm is adopted. The dropout rate is 0.3. The attention dropout rate is 0.1. The activation dropout is 0.1. The initialization learning rate is 1e-07 and the learning rate is 0.0015. The training batch size is 4,096 tokens. We use optimizer Adam with $\beta_1 = 0.9$ and $\beta_2 = 0.98$. We update the gradients for every 2 steps. The number of warmup steps is 8K. For AdaNorm, the hyper-parameter $C$ is set to 2. We average the last 10 checkpoints for evaluation and set the beam size to 5.

**WMT English-German Translation** Following previous work [Vaswani et al., 2017], we use the same dataset splits and the same compound splitting. The pre-processing code is provided by Fairseq. BPE is used to get vocabularies. We use the shared embedding setting and the vocabulary size is 32,765. We use "transformer_wmt_en_de_big_t2t" as our basic model. The setting of PreNorm is adopted. The dropout rate is 0.3. The learning rate is 0.001. The training batch size is 4,096 tokens. We use optimizer Adam with $\beta_1 = 0.9$ and $\beta_2 = 0.98$. The number of warmup steps is 4K. For AdaNorm, the hyper-parameter $C$ is set to 2. We average the last 10 checkpoints for evaluation and set the beam size to 4.

## A.2  Language Modeling

**Enwiki-8**[2] This is a character-level language model dataset with 100M bytes. We use the same preprocessed dataset as in the work [Chung et al., 2017]. We use the code provided by Transformer-XL[3]. We use the default hyper-parameters in the code. The model contains 12 decoder layers and the dimension of each layer is 512. Multi-head attention contains 8 heads and the dimension of each head is 64. The dropout rate is 0.1. The batch size is 22. We use optimizer Adam with a learning rate 0.00025. For AdaNorm, the hyper-parameter $C$ is set to 1. We choose the best checkpoint on the validation set to evaluate the result on the test set.

## A.3  Classification

**RT** The rating inference dataset [Pang and Lee, 2005] is a binary sentiment classification dataset from online movie reviews. Due to the lack of the standard split, we randomly divide all examples into 8,608 for training, 964 for validation, and 1,089 for testing. We implement a 4-layer Transformer encoder. The setting of PreNorm is adopted. The batch size is 4,096 tokens. The word embedding dimension is 128, the hidden dimension is 128. The dropout rate is 0.2. The optimization method is Adam optimizer [Kingma and Ba, 2015] with $\beta_1 = 0.9$, $\beta_2 = 0.998$. For AdaNorm, the hyper-parameter $C$ is set to 0.3.

**SST** The Stanford sentiment treebank [Socher et al., 2013] is a single-sentence classification dataset built on movie reviews. We run experiments on a five label set. It provides the standard spit, with 8,544 for training, 1,101 for validation, and 2,210 for testing. We use the same model structure in RT. For AdaNorm, the hyper-parameter $C$ is set to 0.3. The rest of parameters are set exactly the same as in RT settings.

**MNIST Image Recognition**    The MNIST handwritten digit dataset [LeCun et al., 1998] consists of 55,000 training images, 5,000 validation images, and additional 10,000 testing images. This task aims to recognize the numerical digit (0-9) of each image. We implement a CNN based classifier. The first 2D-convolution layer has 1 in-channel, 20 out-channels. The second 2D-convolution layer has 20 in-channels, 50 out-channels. We flatten the output of the second 2D-convolution layer and send it to a linear layer. The batch size is 32. We use Adam optimizer with a learning rate of $0.001$. We apply LayerNorm before activation in every linear layer. When applying AdaNorm, we set hyper-parameter $C$ to 2. We train the model for 20 epochs. We choose the best checkpoint on the validation set for evaluation.

### A.4   Dependency Parsing

**Transition-based Dependency Parsing** Following previous work, we use English Penn TreeBank (PTB) [Marcus et al., 1993] for experiments. We follow the standard split of the corpus with sections 2-21 as the training set (39,832 sentences, 1,900,056 transition examples), section 22 as the validation set (1,700 sentences, 80,234 transition examples), and section 23 as the testing set (2,416 sentences, 113,368 transition examples). We implement a MLP-based parser following the work [Chen and Manning, 2014]. The dimension of the hidden state is 512, the batch size is $1,024$, the dropout rate is 0.2. We use optimizer Adam and initialize the learning rate to $0.001$. We apply LayerNorm before activation in every linear layer. When applying AdaNorm, we set hyper-parameter $C$ to 1. We train 20 epochs on the training set. We evaluate the model on the development set every epoch and find the best checkpoint to evaluate the test results.

## Footnotes

[1]https://github.com/pytorch/fairseq

[2]http://www.mattmahoney.net/dc/text.html

[3]https://github.com/kimiyoung/transformer-xl

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

# B  Proof of Theorem 1

*Proof.* Define $\mathbf{1}_H = (1, 1, \cdots, 1)^{\mathrm{T}}$. It is easy to verify

$$\mathbf{y}^{\mathrm{T}}\mathbf{y} = \sum_{i=1}^{H} y_i^2 = \sum_{i=1}^{H} (\frac{x_i - \mu}{\sigma_i})^2 = H$$

$$1_H^{\mathrm{T}}\mathbf{1}_H = \sum_{i=1}^{H} 1^2 = H \tag{1}$$

$$\mathbf{y}^{\mathrm{T}}\mathbf{1}_H = \sum_{i=1}^{H} y_i = \sum_{i=1}^{H} \frac{x_i - \mu}{\sigma_i} = 0$$

The forward propagation

$$\mathbf{y} = \frac{\mathbf{x} - \mu\mathbf{1}_H}{\sigma} \tag{2}$$

Calculating the gradient in backward propagation

$$\frac{\partial \mathbf{y}}{\partial \mathbf{x}} = \frac{\partial \big((I\mathbf{x} - \mu\mathbf{1}_H)/\sigma\big)}{\partial \mathbf{x}} = \frac{1}{\sigma}I^{\mathrm{T}} = \frac{1}{\sigma}I$$

$$\frac{\partial \mathbf{y}}{\partial \mu} = \frac{\partial \big((I\mathbf{x} - \mu\mathbf{1}_H)/\sigma\big)}{\partial \mu} = -\frac{1}{\sigma}\mathbf{1}_H^{\mathrm{T}}$$

$$\frac{\partial \mathbf{y}}{\partial \sigma} = \frac{\partial \big((I\mathbf{x} - \mu\mathbf{1}_H)/\sigma\big)}{\partial \sigma} = -\frac{(I\mathbf{x} - \mu\mathbf{1}_H)^{\mathrm{T}}}{\sigma^2} = -\frac{1}{\sigma}\mathbf{y}^{\mathrm{T}}$$

$$\frac{\partial \mu}{\partial x_i} = \frac{\partial \sum_{i=1}^{H} x_i/H}{\partial x_i} = \frac{1}{H} \implies \frac{\partial \mu}{\partial \mathbf{x}} = \frac{\mathbf{1}_H}{H} \tag{3}$$

$$\frac{\partial \sigma}{\partial x_i} = \frac{\partial \sqrt{(\sum_{i=1}^{H} x_i^2 - H\mu^2)/H}}{\partial x_i} = \frac{1}{2\sigma}\frac{\partial(\sigma^2)}{\partial x_i}$$

$$= \frac{1}{2\sigma}\frac{\partial\big((\sum_{i=1}^{H} x_i^2 - H\mu^2)/H\big)}{\partial x_i} = \frac{1}{2\sigma}(2x_i/H - 2\mu\frac{\partial \mu}{\partial x_i})$$

$$= \frac{x_i - \mu}{H\sigma} = \frac{y_i}{H} \implies \frac{\partial \sigma}{\partial \mathbf{x}} = \frac{\mathbf{y}}{H}$$

To conclude

$$\frac{\partial \mathbf{y}}{\partial \mathbf{x}} = \frac{1}{\sigma}I \quad \frac{\partial \mathbf{y}}{\partial \mu} = -\frac{1}{\sigma}\mathbf{1}_H^{\mathrm{T}} \quad \frac{\partial \mathbf{y}}{\partial \sigma} = -\frac{1}{\sigma}\mathbf{y}^{\mathrm{T}} \quad \frac{\partial \mu}{\partial \mathbf{x}} = \frac{\mathbf{1}_H}{H} \quad \frac{\partial \sigma}{\partial \mathbf{x}} = \frac{\mathbf{y}}{H} \tag{4}$$

If we detach the gradient of $\mu$ and $\sigma$, in backward propagation

$$\frac{\partial \ell}{\partial \mathbf{x}} = \frac{\partial \mathbf{y}}{\partial \mathbf{x}}\frac{\partial \ell}{\partial \mathbf{y}} = \frac{1}{\sigma}\frac{\partial \ell}{\partial \mathbf{y}} \tag{5}$$

namely

$$a_i = \frac{g_i}{\sigma} \tag{6}$$

Calculating $\bar{a}$ and $D_a$

$$H\bar{a} = \sum_{i=1}^{H} a_i = \sum_{i=1}^{H} \frac{g_i}{\sigma} = \frac{H\bar{g}}{\sigma}$$

$$HD_a = \sum_{i=1}^{H} (a_i - \bar{a})^2 = \sum_{i=1}^{H} (\frac{g_i - \bar{g}}{\sigma})^2 = \frac{HD_g}{\sigma^2} \tag{7}$$

To conclude, $\bar{a} = \bar{g}/\sigma$ and $D_a = D_g/(\sigma^2)$.

**Proof of (1)**

(1) In standard layernorm, we do not detach the gradients of $\mu$ and $\sigma$, in backward propagation

$$\frac{\partial \ell}{\partial \mathbf{x}} = (\frac{\partial \mathbf{y}}{\partial \mathbf{x}} + \frac{\partial \mu}{\partial \mathbf{x}}\frac{\partial \mathbf{y}}{\partial \mu} + \frac{\partial \sigma}{\partial \mathbf{x}}\frac{\partial \mathbf{y}}{\partial \sigma})\frac{\partial \ell}{\partial \mathbf{y}} = \frac{1}{\sigma}(I - \frac{\mathbf{y}\mathbf{y}^{\mathrm{T}}}{H} - \frac{\mathbf{1}_H \mathbf{1}_H^{\mathrm{T}}}{H})\frac{\partial \ell}{\partial \mathbf{y}} \tag{8}$$

Define $W_1 = \frac{1}{\sigma}(I - \frac{\mathbf{y}\mathbf{y}^{\mathrm{T}}}{H} - \frac{\mathbf{1}_H \mathbf{1}_H^{\mathrm{T}}}{H})$, we can verify that

$$\mathbf{1}_H^{\mathrm{T}} W_1 = \mathbf{1}_H^{\mathrm{T}}\frac{1}{\sigma}(I - \frac{\mathbf{1}_H \mathbf{1}_H^{\mathrm{T}} + \mathbf{y}\mathbf{y}^{\mathrm{T}}}{H}) = \frac{1}{\sigma}(\mathbf{1}_H - \frac{\mathbf{1}_H^{\mathrm{T}}\mathbf{1}_H}{H}\mathbf{1}_H^{\mathrm{T}} - \frac{\mathbf{1}_H^{\mathrm{T}}\mathbf{y}}{H}\mathbf{y}^{\mathrm{T}}) = \frac{\mathbf{1}_H - \mathbf{1}_H - 0}{\sigma} = 0 \tag{9}$$

Therefore,

$$H\bar{b} = \sum_{i=1}^{H} b_i = \mathbf{1}_H^{\mathrm{T}}(b_1, b_2, ..., b_H)^{\mathrm{T}} = \mathbf{1}_H^{\mathrm{T}} W_1 (g_1, g_2, ..., g_H)^{\mathrm{T}} = 0 \tag{10}$$

For any vector $\mathbf{u}$ vertical to $\mathbf{1}_H$ and $\mathbf{y}$ ( $\mathbf{1}_H$ is vertical to $\mathbf{y}$), we have

$$W_1 \mathbf{u} = \frac{1}{\sigma}(I - \frac{\mathbf{1}_H \mathbf{1}_H^{\mathrm{T}} + \mathbf{y}\mathbf{y}^{\mathrm{T}}}{H})\mathbf{u} = \frac{1}{\sigma}(\mathbf{u} - \mathbf{1}_H \frac{\mathbf{1}_H^{\mathrm{T}}\mathbf{u}}{H} - \mathbf{y}\frac{\mathbf{y}^{\mathrm{T}}\mathbf{u}}{H}) = \frac{1}{\mathbf{u} - 0 - 0} = \frac{1}{\sigma}\mathbf{u}$$

$$W_1 \mathbf{y} = \frac{1}{\sigma}(I - \frac{\mathbf{1}_H \mathbf{1}_H^{\mathrm{T}} + \mathbf{y}\mathbf{y}^{\mathrm{T}}}{H})\mathbf{y} = \frac{1}{\sigma}(\mathbf{y} - \mathbf{1}_H \frac{\mathbf{1}_H^{\mathrm{T}}\mathbf{y}}{H} - \mathbf{y}\frac{\mathbf{y}^{\mathrm{T}}\mathbf{y}}{H}) = \frac{\mathbf{y} - 0 - \mathbf{y}}{\sigma} = 0 \tag{11}$$

$$W_1 \mathbf{1}_H = \frac{1}{\sigma}(I - \frac{\mathbf{1}_H \mathbf{1}_H^{\mathrm{T}} + \mathbf{y}\mathbf{y}^{\mathrm{T}}}{H})\mathbf{1}_H = \frac{1}{\sigma}(\mathbf{1}_H - \mathbf{1}_H \frac{\mathbf{1}_H^{\mathrm{T}}\mathbf{1}_H}{H} - \mathbf{y}\frac{\mathbf{y}^{\mathrm{T}}\mathbf{1}_H}{H}) = \frac{\mathbf{1}_H^{\mathrm{T}} - \mathbf{1}_H^{\mathrm{T}} - 0}{\sigma} = 0$$

We expand $\mathbf{1}_H$ and $\mathbf{y}$ to a standard orthogonal basis $\mathbf{u}_1 = \mathbf{1}_H/\sqrt{H}, \mathbf{u}_2 = \mathbf{y}/\sqrt{H}, \mathbf{u}_3, ..., \mathbf{u}_H$, then for any vector $\mathbf{v} = \sum_{i=1}^{H} \lambda_i \mathbf{u}_i$, we have

$$W_1 \mathbf{v} = \sum_{i=1}^{H} \lambda_i W_1 \mathbf{u}_i = W_1 \mathbf{1}_H/\sqrt{H} + W_1 \mathbf{y}/\sqrt{H} + \sum_{i=3}^{H} \lambda_i W_1 \mathbf{u}_i = \frac{1}{\sigma}\sum_{i=3}^{H} \lambda_i \mathbf{u}_i$$

$$\|W_1 \mathbf{v}\|^2 = \frac{1}{\sigma^2}\sum_{i=3}^{H} \lambda_i^2 \leq \frac{1}{\sigma^2}\sum_{i=1}^{H} \lambda_i^2 = \frac{1}{\sigma^2}\|\mathbf{v}\|^2 \tag{12}$$

Therefore,

$$
\begin{aligned}
HD_b &= \sum_{i=1}^{H}(b_i - \bar{b})^2 \\
&= \sum_{i=1}^{H} b_i^2 \\
&= \|(b_1, b_2, ..., b_H)^{\mathrm{T}}\|^2 \\
&= \|W_1(g_1, g_2, ..., g_H)^{\mathrm{T}}\|^2 \\
&= \|W_1(g_1 - \bar{g}, g_2 - \bar{g}, ..., g_H - \bar{g})^{\mathrm{T}} + W_1 \bar{g}\mathbf{1}_H\|^2 \\
&= \|W_1(g_1 - \bar{g}, g_2 - \bar{g}, ..., g_H - \bar{g})^{\mathrm{T}}\|^2 \\
&\leq \frac{1}{\sigma^2}\|(g_1 - \bar{g}, g_2 - \bar{g}, ..., g_H - \bar{g})^{\mathrm{T}}\|^2 \\
&= \frac{1}{\sigma^2}\sum_{i=1}^{H}(g_i - g)^2 \quad \text{(Ineq. 12)} \\
&= \frac{1}{\sigma^2}HD_g \\
&= HD_a
\end{aligned}
\tag{13}
$$

To conclude, $\bar{b} = 0$ and $D_b \leq D_a = D_g/(\sigma^2)$.

**Proof of (2)** (2) If we detach the gradients of $\mu$, in backward propagation

$$\frac{\partial \ell}{\partial \mathbf{x}} = \left(\frac{\partial \mathbf{y}}{\partial \mathbf{x}} + \frac{\partial \sigma}{\partial \mathbf{x}}\frac{\partial \mathbf{y}}{\partial \sigma}\right)\frac{\partial \ell}{\partial \mathbf{y}} = \frac{1}{\sigma}\left(I - \frac{\mathbf{y}\mathbf{y}^{\mathrm{T}}}{H}\right)\frac{\partial \ell}{\partial \mathbf{y}} \tag{14}$$

Define $W_2 = \frac{1}{\sigma}(I - \frac{\mathbf{y}\mathbf{y}^{\mathrm{T}}}{H})$, then

$$\mathbf{1}_H^{\mathrm{T}} W_2 = \mathbf{1}_H^{\mathrm{T}} \frac{1}{\sigma}\left(I - \frac{\mathbf{y}\mathbf{y}^{\mathrm{T}}}{H}\right) = \frac{1}{\sigma}\left(\mathbf{1}_H^{\mathrm{T}} - \frac{\mathbf{1}_H^{\mathrm{T}}\mathbf{y}}{H}\mathbf{y}^{\mathrm{T}}\right) = \frac{\mathbf{1}_H^{\mathrm{T}} - 0}{\sigma} = \frac{\mathbf{1}_H^{\mathrm{T}}}{\sigma} \tag{15}$$

Therefore,

$$H\bar{c} = \sum_{i=1}^{H} c_i = \mathbf{1}_H^{\mathrm{T}}(c_1,...,c_H)^{\mathrm{T}} = \mathbf{1}_H^{\mathrm{T}} W_2 (g_1,...,g_H)^{\mathrm{T}} = \frac{\mathbf{1}_H^{\mathrm{T}}(g_1,...,g_H)^{\mathrm{T}}}{\sigma} = \sum_{i=1}^{H} g_i = \frac{H\bar{g}}{\sigma} \tag{16}$$

Consider

$$\begin{aligned}
(I - \mathbf{1}_H\mathbf{1}_H^{\mathrm{T}}/H)W_2 &= (I - \mathbf{1}_H\mathbf{1}_H^{\mathrm{T}}/H)\frac{1}{\sigma}(I - \mathbf{y}\mathbf{y}^{\mathrm{T}}/H) \\
&= \frac{1}{\sigma}(I - \mathbf{1}_H\mathbf{1}_H^{\mathrm{T}}/H - \mathbf{y}\mathbf{y}^{\mathrm{T}}/H + \mathbf{1}_H\mathbf{1}_H^{\mathrm{T}}\mathbf{y}\mathbf{y}^{\mathrm{T}}/(H^2)) \\
&= \frac{1}{\sigma}(I - \mathbf{1}_H\mathbf{1}_H^{\mathrm{T}}/H - \mathbf{y}\mathbf{y}^{\mathrm{T}}/H + 0) \\
&= W_1
\end{aligned} \tag{17}$$

Therefore,

$$\begin{aligned}
HD_c &= \sum_{i=1}^{H}(c_i - \bar{c})^2 \\
&= \|(c_1 - \bar{c}, c_2 - \bar{c}, ..., c_H - \bar{c})^{\mathrm{T}}\|^2 \\
&= \|(I - \mathbf{1}_H\mathbf{1}_H^{\mathrm{T}}/H)(c_1, c_2, ..., c_H)^{\mathrm{T}}\|^2 \\
&= \|(I - \mathbf{1}_H\mathbf{1}_H^{\mathrm{T}}/H)W_2(g_1, g_2, ..., g_H)^{\mathrm{T}}\|^2 \\
&= \|W_1(g_1 - \bar{g}, g_2 - \bar{g}, ..., g_H - \bar{g})^{\mathrm{T}}\|^2 \\
&\leq \frac{1}{\sigma^2}\|(g_1 - \bar{g}, g_2 - \bar{g}, ..., g_H - \bar{g})^{\mathrm{T}}\|^2 \\
&= \frac{1}{\sigma^2}\sum_{i=1}^{H}(g_i - \bar{g})^2 \quad \text{(Ineq. 12)} \\
&= \frac{1}{\sigma^2}HD_g \\
&= HD_a
\end{aligned} \tag{18}$$

To conclude, $\bar{c} = \bar{a} = \bar{g}/\sigma$, $D_c \leq D_a = D_g/(\sigma)^2$.

**Proof of (3)**

(3) If we detach the gradient of $\sigma$, in backward propagation

$$\frac{\partial \ell}{\partial \mathbf{x}} = \left(\frac{\partial \mathbf{y}}{\partial \mathbf{x}} + \frac{\partial \mu}{\partial \mathbf{x}}\frac{\partial \mathbf{y}}{\partial \mu}\right)\frac{\partial \ell}{\partial \mathbf{y}} = \frac{1}{\sigma}\left(I - \frac{\mathbf{1}_H\mathbf{1}_H^{\mathrm{T}}}{H}\right)\frac{\partial \ell}{\partial \mathbf{y}} \tag{19}$$

Define $W_3 = \frac{1}{\sigma}(I - \frac{\mathbf{1}_H\mathbf{1}_H^{\mathrm{T}}}{H})$, we can verify that

$$\mathbf{1}_H^{\mathrm{T}} W_3 = \mathbf{1}_H^{\mathrm{T}} \frac{1}{\sigma}\left(I - \frac{\mathbf{1}_H\mathbf{1}_H^{\mathrm{T}}}{H}\right) = \frac{1}{\sigma}\left(\mathbf{1}_H^{\mathrm{T}} - \frac{\mathbf{1}_H^{\mathrm{T}}\mathbf{1}_H}{H}\mathbf{1}_H^{\mathrm{T}}\right) = \frac{\mathbf{1}_H^{\mathrm{T}} - \mathbf{1}_H^{\mathrm{T}}}{\sigma} = 0 \tag{20}$$

Therefore,

$$H\bar{d} = \sum_{i=1}^{H} d_i = \mathbf{1}_H^{\mathrm{T}}(d_1, d_2, ..., d_H)^{\mathrm{T}} = \mathbf{1}_H^{\mathrm{T}} W_3(g_1, g_2, ..., g_H)^{\mathrm{T}} = 0 \tag{21}$$

For any vector $\mathbf{u}$ vertical to $\mathbf{1}_H$

$$W_3 \mathbf{1}_H = \frac{1}{\sigma}(I - \frac{\mathbf{1}_H \mathbf{1}_H^{\mathrm{T}}}{H})\mathbf{1}_H = \frac{1}{\sigma}(\mathbf{1}_H - \frac{\mathbf{1}_H \mathbf{1}_H^{\mathrm{T}} \mathbf{1}_H}{H}) = \frac{\mathbf{1}_H - \mathbf{1}_H}{\sigma} = 0$$

$$W_3 \mathbf{u} = \frac{1}{\sigma}(I - \frac{\mathbf{1}_H \mathbf{1}_H^{\mathrm{T}}}{H})\mathbf{u} = \frac{1}{\sigma}(\mathbf{u} - \frac{\mathbf{1}_H \mathbf{1}_H^{\mathrm{T}} \mathbf{u}}{H}) = \frac{\mathbf{u} - 0}{\sigma} = \frac{\mathbf{u}}{\sigma} \tag{22}$$

Note that $(g_1 - \bar{g}, g_2 - \bar{g}, ..., g_H - \bar{g})\mathbf{1}_H = \sum_{i=1}^{H}(g_i - \bar{g}) = 0$, namely $(g_1 - \bar{g}, g_2 - \bar{g}, ..., g_H - \bar{g})^{\mathrm{T}}$ is vertical to $\mathbf{1}_H$ and $W_3(g_1 - \bar{g}, g_2 - \bar{g}, ..., g_H - \bar{g})^{\mathrm{T}} = (g_1 - \bar{g}, g_2 - \bar{g}, ..., g_H - \bar{g})^{\mathrm{T}}$, therefore

$$
\begin{aligned}
HD_d &= \sum_{i=1}^{H}(d_i - \bar{d})^2 \\
&= \sum_{i=1}^{H}(d_i)^2 \\
&= \|(d_1, d_2, ..., d_H)^{\mathrm{T}}\|^2 \\
&= \|W_3(g_1, g_2, ..., g_H)^{\mathrm{T}}\|^2 \\
&= \|W_3(g_1 - \bar{g}, g_2 - \bar{g}, ..., g_H - \bar{g})^{\mathrm{T}} + W_3 \bar{g} \mathbf{1}_H\|^2 \\
&= \|W_3(g_1 - \bar{g}, g_2 - \bar{g}, ..., g_H - \bar{g})^{\mathrm{T}}\|^2 \\
&= \|(g_1 - \bar{g}, g_2 - \bar{g}, ..., g_H - \bar{g})^{\mathrm{T}}\|^2 \\
&= \frac{1}{\sigma^2}\sum_{i=1}^{H}(g_i - g)^2 \\
&= \frac{1}{\sigma^2}HD_g \\
&= HD_a
\end{aligned}
\tag{23}
$$

To concluede $\bar{d} = 0$ and $D_d = D_a = D_g/(\sigma^2)$.

$\square$

# C   Proof of Theorem 2

*Proof.* Assume $d\mathbf{y} = (dy_1, dy_2, ..., dy_H)$. Because $\phi$ is derivable, asuume $\mathbf{v} = (\phi'(y_1), \phi'(y_2), ..., \phi'(y_H))$. It is easy to verify

$$\sum_{i=1}^{H} \phi(y_i) = HC$$
$$\sum_{i=1}^{H} y_i = 0 \qquad (24)$$
$$\sum_{i=1}^{H} y_i^2 = H$$

Differential on both sides of following three equations

$$\mathbf{v}^{\mathsf{T}} d\mathbf{y} = \sum_{i=1}^{H} \phi'(y_i) dy_i = 0 \qquad (25)$$

$$\mathbf{1}_H^{\mathsf{T}} d\mathbf{y} = \sum_{i=1}^{H} dy_i = 0 \qquad (26)$$

$$\mathbf{y}^{\mathsf{T}} d\mathbf{y} = \sum_{i=1}^{H} y_i dy_i = 0 \qquad (27)$$

In $H$-dim Euclidean space, note that $\mathbf{y}^{\mathsf{T}} \mathbf{1}_H = \sum_{i=1}^{H} y_i = 0$, namely $\mathbf{1}_H$ and $\mathbf{y}$ are vertical. We expand $\mathbf{1}_H$ and $\mathbf{y}$ to an orthogonal basis $\mathbf{u}_1 = \mathbf{1}_H, \mathbf{u}_2 = \mathbf{y}, \mathbf{u}_3, ..., \mathbf{u}_H$. Suppose $d\mathbf{y} = \sum_{i=1}^{H} \alpha \mathbf{u}_i$ and $\mathbf{v} = \sum_{i=1}^{H} \beta_i \mathbf{u}_i$, we have

$$\mathbf{v}^{\mathsf{T}} d\mathbf{y} = \sum_{i=1}^{H} \alpha_i \beta_i \|\mathbf{u}_i\|^2 = 0$$
$$\mathbf{1}_H^{\mathsf{T}} d\mathbf{y} = \alpha_1 \|\mathbf{u}_1\|^2 = 0 \qquad (28)$$
$$\mathbf{y}^{\mathsf{T}} d\mathbf{y} = \alpha_2 \|\mathbf{u}_2\|^2 = 0$$

Accoring to Eq. 28, $\sum_{i=3}^{H} \alpha_i \beta_i \|\mathbf{u}_i\|^2 = 0$. Because it holds in spite of $\alpha_i, (i > 2)$, $\beta_i = 0, (i > 2)$. Therefore, $\mathbf{v} = \beta_1 \mathbf{1}_H + \beta_2 \mathbf{y}$. Namely

$$\phi'(y_i) = \beta_1 + \beta_2 y_i$$
$$\phi(y_i) = C_1 + C_2 y_i + C_3 y_i^2 \qquad (29)$$

Consider $z_i$

$$M > |\frac{1}{H}\sum_{i=1}^{H}z_i|$$

$$= |\frac{1}{H}\sum_{i=1}^{H}\phi(y_i)y_i|$$

$$= |\frac{1}{H}\sum_{i=1}^{H}C_1y_i + C_2y_i^2 + C_3y_i^3|$$

$$= |\frac{1}{H}(C_1\sum_{i=1}^{H}y_i + C_2\sum_{i=1}^{H}y_i^2 + C_3\sum_{i=1}^{H}y_i^3)| \tag{30}$$

$$= |\frac{1}{H}(C_2H + C_3\sum_{i=1}^{H}y_i^3)|$$

$$= |C_2 + \frac{C_3}{H}\sum_{i=1}^{H}y_i^3|$$

$$\geq |\frac{C_3}{H}\sum_{i=1}^{H}y_i^3| - |C_2|$$

If we set $y_1 = 2\sqrt{H/6}, y_2 = y_3 = -\sqrt{H/6}, y_i = 0$ $(i > 3)$, then $\sum_{i=1}y_i = 0$ and $\sum_{i=1}y_i^2 = \frac{4H}{6} + \frac{H}{6} + \frac{4H}{6} = H$ hold. Under this circumstances

$$M > |\frac{C_3}{H}\sum_{i=1}^{H}y_i^3| - |C_2|$$

$$= |\frac{C_3}{H}((2\sqrt{H/6})^3 + (-\sqrt{H/6})^3) + (-\sqrt{H/6})^3)| - |C_2| \tag{31}$$

$$= |\frac{C_3}{H}(\frac{4H}{3} - \frac{H}{6} - \frac{H}{6})\sqrt{H/6}| - |C_2|$$

$$= |C_3\sqrt{H/6}| - |C_2|$$

Therefore $|C_3| < \frac{|C_2|+M}{\sqrt{H/6}}$ holds for any $H$, when $H$ approches infinity, we have $|C_3| = 0$. $\sum_{i=1}^{H}\phi(y_i) = C_1H + C_2(\sum_{i=1}^{H}y_i) = C_1H = CH$, therefore $C_1 = C$. Let $k = -C_2/C$, therefore $\phi(y_i) = C(1 - ky_i)$, then

$$M > |\frac{1}{H}\sum_{i=1}^{H}z_i|$$

$$= |\frac{1}{H}\sum_{i=1}^{H}\phi(y_i)y_i|$$

$$= |\frac{1}{H}\sum_{i=1}^{H}C_1y_i + C_2y_i^2 + C_3y_i^3| \tag{32}$$

$$= |\frac{1}{H}(C\sum_{i=1}^{H}y_i - k\sum_{i=1}^{H}y_i^2)|$$

$$= |k|$$

Namely $M > |\frac{1}{H}\sum_{i=1}^{H}z_i|$ can hold if $M > |k|$. To conclude $\phi(y_i) = C(1 - ky_i)$ is the only solution. $\qquad\square$