[Reviews · NeurIPS 2019]

Reviewer 1



Originality: Though the analysis is new, the proposed algorithm is not. Quality: To motivate AdaNorm, DetachNorm is first used for ablation study. Significance: Give the minor empirical results, it seem hard for the practitioners to widely adopt AdaNorm.

Reviewer 2



The paper presents a simple yet an effective idea that is based on a rigorous analysis on the effects of the bias and gain used in LayerNorm. The paper is written very clearly. The experimental setup is excellent, and all the details on the model size, hyper-parameters are described in the supplementary material. There is a typo in the table 2: imporvement -> improvement

Reviewer 3



The authors explore two ideas in this work: 1) analysis of different parts of LayerNorm (normalization itself and gradients); 2) new adaptive LayerNorm where bias and gain of LayerNorm are replaced by a new adaptive scaling approach. The authors provide a theoretical analysis of both parts, and provide empirical analysis of the discussed ideas that explain and justify the two topics that are explored. Detailed comments are given below: - The title is very vague, and should be modified/expanded to better convey the topics being explored. - As mentioned above, the paper consists of two somewhat unrelated topics. Although the authors attempt to connect them, it is not really clear that they do belong under one roof. - Equation for variance in (1) seems wrong. Is that the equation that was used or is it a typo? I tend to think the former as it repeats several times throughout the paper. - DetachNorm does seem interesting, but it could be that poor results are (at least to a degree) due to a fact that the model parameters are not being trained to optimize the intended loss as the gradient is simply wrong (due to parts of the gradient being detached and essentially random noise is added into the model through the special copy function). Can you please elaborate on this, maybe I am missing something? - I am not sure that (3) is necessary to add to the paper as it is most likely clear to the readers already. But it is OK to leave it if the authors see fit. - Table 2 could and should be consolidated as it basically repeats the same info twice. - At the start of Section 4 the authors say "The reason for overfitting is probably ...", where the reason is not really shown previously. Seems like a guess. - Also, the proposed AdaNorm does not really directly address the items discussed in the first part of the paper. The two topics seem to be glued together. - "Unlike bias and gain being fixed ...", bias and gain have not actually been shown anywhere in the paper. Would be good to remove some of the unnecessary equations and add some that would actually help a reader follow the text. - In Theorem 2 and above, should the absolute value be only around z_i and not the entire sum? What if z_i's are large but they cancel each other out? - Some notation is used without being introduced, such as E below Theorem 2. This should be fixed. - "To prevent ... dismissing the feature of gradient", what does this even mean? This is just given very briefly, but it seems that it should be much better explained. All in all it is an interesting paper, but a number of critical parts are not well explained and should be much better elaborated. ### After reading the authors' feedback ### Thank you for the feedback. Not all the comment were addressed which is not ideal, but could be sufficient given limited space. I am increasing my recommendation to "weak accept", as I don't believe that the paper in its current state warrants higher than that.

[Author Response · NeurIPS 2019]

**To all reviwers:**

We thank the reviewers for their detailed comments. The followings are our responses.

**To reviewer 1:**

[1] Are the empirical improvements strong?

We have strong confidence in its empirical improvements. Take the results on translation tasks as an example (shown in Table 1). AdaNorm has brought improvements of 0.6 BLEU on En-Vi and 0.8 BLEU on De-En, much higher than other techniques do like Fixup and KD.

[2] More intuitions for which task AdaNorm can really improve, and why.

AdaNorm works better for tasks requiring complex model structures. The reason is that deeper models usually have the tendency to over-fit training data, and AdaNorm alleviates the over-fitting by adaptively controlling scaling weights towards different inputs on affine transformation. Comparing to LayerNorm that ignores the input distribution when testing, our proposed AdaNorm has achieved better empirical improvements.

**To reviewers 2:**

Thanks for your comments and suggestions.

**To reviewer 3:**

[1] Equation for variance in (1) seems wrong.

The equation for variance in (1) is correct. It is a variant of the traditional variance equation. The followings are the derivation process. If $\sigma^2$ is the variance of $X$, then

$$\sigma^2 = \mathrm{E}[(X - \mathrm{E}[X])^2] = \mathrm{E}[X^2 - 2X\,\mathrm{E}[X] + \mathrm{E}[X]^2] = \mathrm{E}[X^2] - 2\,\mathrm{E}[X]\,\mathrm{E}[X] + \mathrm{E}[X]^2 = \mathrm{E}[X^2] - \mathrm{E}[X]^2 \tag{1}$$

where $E$ is a mean function. In this paper, $X = x_1, x_2, \cdots, x_H$ and the variance can be written as $\sigma^2 = \frac{1}{n}(\sum_{i=1}^n x_i^2 - n\mu^2)$.

[2] In DetachNorm, the gradient is simply wrong (due to parts of the gradient being detached and essentially random noise is added into the model through the special copy function).

Here we illustrate its correctness by analyzing the two mentioned operations. First the detaching operation simulates the situation of constant variance and mean that have zero gradient to the input. Comparing to LayerNorm, they are two settings to evaluate the effect of variance and mean on gradients. The gradients in these two settings are different, but they are both right. Second, the special copy function is a simple assignment operation. It has extremely weak effect on model performance considering the huge amount of assignment operations in neural networks.

[3] The proposed AdaNorm does not really directly address the items discussed in the first part of the paper.

As described in lines 193-197, AdaNorm is proposed to address the over-fitting problem discussed in the first part. The first part analyzes which parts in LayerNorm work and which parts do not. Empirical results show that "bias and gain", parameters of LayerNorm, are not always beneficial because they increase the risk of over-fitting. Motivated by this fact, we propose a new normalization approach, AdaNorm, to address the over-fitting problem. Experiment results demonstrate that AdaNorm outperforms LayerNorm on seven datasets with better convergence.

[4] In Theorem 2 and above, should the absolute value be only around $z_i$ and not the entire sum. What if $z_i$ are large but they cancel each other out?

Thanks for your suggestions. We will consider replacing $|\sum_{i=1}^{H} z_i|/H < M$ with $\sum_{i=1}^{H} |z_i|/H < M$. For the proof of the theorem, we only need $|\sum_{i=1}^{H} z_i|/H < M$. Since $\sum_{i=1}^{H} |z_i|/H < M$ is a stronger constraint, it does not affect the proof.

[5] "To prevent ... dismissing the feature of gradient", what does this even mean?

It means that LayerNorm has an advantage of re-centering and re-scaling gradients. The proposed AdaNorm still keeps this advantage when avoiding the over-fitting problem.

| Models | En-Vi | De-En |
|---|---|---|
| NPMT+LM (Huang et al., 2017) | 28.1 | 30.1 |
| Risk (Edunov et al., 2018) | - | 32.8 |
| Var-Attn (Deng et al., 2018) | - | 33.7 |
| Transformer | 30.1 | 34.2 |
| +KD (Tan et al., 2018) | 28.7 | 34.0 |
| +Fixup (Zhang et al., 2019) | - | 34.5 |
| +AdaNorm | **30.7** | **35.0** |

Table 1: Results on the IWSLT15 English-to-Vietnamese translation test set and IWSLT14 German-to-English test set.



[Meta-Review · NeurIPS 2019]

The paper presents an extension to layer norm and provides theoretical analysis to the normalization scheme. There was concern about the direct applicability of the proposed AdaNorm, but general consensus is that the paper adds an interesting contribution to the understanding of LayerNorm related methods.